- **Profiling aerosol optical, microphysical and hygroscopic**
- properties in ambient conditions by combining in-situ and
- remote sensing
- Alexandra Tsekeri<sup>1</sup>, Vassilis Amiridis<sup>1</sup>, Franco Marenco<sup>2</sup>, Athanasios Nenes<sup>3,4,5</sup>,
- Eleni Marinou<sup>1,6</sup>, Stavros Solomos<sup>1</sup>, Phil Rosenberg<sup>7</sup>, Jamie Trembath<sup>8</sup>, Graeme
- 7 J. Nott<sup>8</sup>, James Allan<sup>9,10</sup>, Michael Le Breton<sup>9</sup>, Asan Bacak<sup>9</sup>, Hugh Coe<sup>9</sup>, Carl
- Percival<sup>9</sup>, and Nikolaos Mihalopoulos<sup>4</sup>
- [1]{IAASARS, National Observatory of Athens, Athens, Greece}
- [2]{Satellite Applications, Met Office, Exeter, UK}
- [3]{School of Earth and Atmospheric Sciences and Chemical and Biomolecular Engineering,
- Georgia Institute of Technology, Atlanta, Georgia, USA}
- [4]{IERSD, National Observatory of Athens, Athens, Greece}
- [5]{ICE-HT, Foundation for Research and Technology Hellas, Patras, Greece}
- [6]{Laboratory of Atmospheric Physics, Aristotle University of Thessaloniki, Thessaloniki,Greece}
- [7] {School of Earth and Environment, University of Leeds, Leeds, UK}
- [8] {Facility for Airborne Atmospheric Measurements (FAAM), Cranfield, UK}
- [9]{School of Earth, Atmospheric and Environmental Sciences, University of Manchester,
- Manchester, UK}
- [10]{National Centre for Atmospheric Science, The University of Manchester, Manchester,
- UK}
- Correspondence to: A. Tsekeri (atsekeri@noa.gr)

#### 1 Abstract

2 We present the In-situ/Remote sensing aerosol Retrieval Algorithm (IRRA) that combines 3 airborne in-situ and lidar remote sensing data to retrieve vertical profiles of ambient aerosol optical, microphysical and hygroscopic properties, employing the ISORROPIA II model for 4 5 acquiring the hygroscopic growth. Here we apply the algorithm on data collected from the 6 Facility for Airborne Atmospheric Measurements (FAAM) BAe-146 research aircraft during 7 the ACEMED campaign in Eastern Mediterranean: vertical profiles of aerosol microphysical 8 properties have been derived successfully for an aged smoke plume near the city of Thessaloniki 9 with typical lidar ratios of  $\sim$ 60-80 sr at 532 nm, along with single scattering albedos of  $\sim$ 0.9-10 0.95 at 550 nm. The aerosol layer reaches the 3.5 km with aerosol optical depth at  $\sim$ 0.4 at 532 11 nm. Our analysis shows that the smoke particles are highly hydrated above land, with 55% and 12 80% water volume content for ambient relative humidity of 80% and 90%, respectively. The 13 proposed methodology is highly advantageous for aerosol characterization in humid conditions 14 and can find valuable applications in aerosol-cloud interaction schemes. Moreover, it can be 15 used for the validation of active space-borne sensors, as is demonstrated here for the case of 16 CALIPSO.

17

#### 18 **1** Introduction

19 Liquid water is by far the most abundant species found in atmospheric particulates, being on 20 average 2-3 times the total aerosol dry mass on a global average (e.g. Pilinis et al., 1995; Liao 21 and Seinfeld, 2005). Aerosol water uptake changes the particle size and refractive index with 22 profound implications for radiative transfer and cloud formation (e.g., Quinn et al., 2005). For 23 example, at a Relative Humidity (RH) of 90%, the scattering cross section can increase by a 24 factor of 5 compared to that of the dry particle (Malm and Day, 2001). On that account, the 25 particle liquid water uptake may greatly affect the aerosol direct radiative cooling (Pilinis et al., 1995; Hegg et al., 1997), currently estimated to range between -0.95 and +0.05 W m<sup>-2</sup> (IPCC, 26 27 2013).

Acquiring the hydrated particle properties is far from trivial, especially when it comes to vertical profiling. In-situ techniques can provide vertically-resolved information when applied by an airborne platform, a solution that is both costly and sparse over space and time. Moreover, the commonly used techniques are subject to limitations, since their application can cause alterations in the particle ambient state even when minimally-invasive instruments are used

(e.g. open-path optical sensors; Snider and Petters, 2008). To address these biases, ambient
 particle samples are first dried and then rehydrated in the controlled environment of an in-situ
 sensor; aerosol properties and changes thereof are then used to understand the behaviour of
 ambient aerosol for any meteorological state (Engelhart et al., 2011; Pikridas et al., 2012).

5 In contrast to in-situ techniques, remote sensing is not invasive and may sample large 6 atmospheric volumes allowing an unprecedented global aerosol monitoring. Passive remote 7 sensing techniques provide columnar particle properties, while active sensors can provide 8 vertically-resolved properties. A well-known active remote sensing instrument is the lidar 9 (Light Detection and Ranging), a sensor that is capable of providing vertical profiles of the 10 backscatter and extinction coefficients at one or more wavelengths. Unfortunately, due to the 11 limited information in lidar measurement content, the ill-posed nature of the aerosol property 12 retrieval remains the inherent disadvantage of the lidar technique, although considerable 13 hardware and algorithmic developments have been achieved over the last decade. These include 14 for example the employment of sophisticated multi-wavelength elastic/Raman lidar 15 measurements in lidar stand-alone retrievals (e.g. Müller et al., 2015), or the combination of 16 elastic lidar with sunphotometer measurements (e.g. Chaikovsky et al, 2015; Lopatin et al., 17 2013). Although these advancements have provided the means for accurate aerosol profiling, 18 the lidar stand-alone retrievals work well only for fine particles while the lidar/sunphotometer 19 retrievals do not fully resolve the particle microphysical property profiles; they rather provide 20 only the particle concentration profile and consider a constant size distribution and refractive 21 index for the whole atmospheric column.

22 An alternative hybrid approach for obtaining well-constrained ambient aerosol profiling is 23 through the utilization of the synergy of active remote sensing observations with concurrent in-24 situ measurements. To date, most efforts towards this direction have focused on low-humidity 25 profiles, so that the dry in-situ measurements refer to ambient particles (e.g. Weinzierl et. al, 26 2009). High-humidity conditions have also been studied, but only for fine mode particle 27 properties (e.g. Ziemba et al. 2013), as the coarse particle hygroscopic growth is not easily 28 constrained with in-situ airborne techniques, mainly due to inlet loses. The IRRA approach 29 presented here addresses these limitations through a combination of in-situ, active remote 30 sensing and hygroscopic modelling, making possible the vertical profiling of fine and coarse 31 particles even for humid conditions. For this purpose, the retrieval combines typical airborne 32 in-situ instrumentation, measuring the dry particle size distribution and chemical composition,

- together with a simple backscatter lidar. The ambient remote sensing measurements are linked
- to the dry in-situ data through modelling of the particle hygroscopic growth with ISORROPIA
- II model (Fountoukis and Nenes, 2007; Guo et al., 2015).
- In the current study IRRA is applied on data collected in the framework of the EUFAR-
- ACEMED campaign ("evaluation of CALIPSO's Aerosol Classification scheme over Eastern
- MEDiterranean"), during which the FAAM BAe-146 research aircraft performed two under-
- flights of the Cloud–Aerosol Lidar and Infrared Pathfinder Satellite Observation (CALIPSO)
- satellite. The Mediterranean is considered ideal for the application and evaluation of IRRA
- retrieval scheme, as almost all globally-relevant aerosol types are encountered in the region, i.e.
  dust storms from desert or semi-arid areas in Africa, fresh and aged smoke from biomass
  burning, maritime aerosols, biogenic emissions, and anthropogenic aerosols (e.g. Lelieveld et
- al., 2002).

In the following sections, the IRRA methodology is presented in section 2, along with a detailed description of the airborne in-situ and lidar measurements acquired during the ACEMED campaign, as well as the ISORROPIA II and other models used. Section 3 presents the IRRA results for the ACEMED flight over Thessaloniki, Greece, along with a comparison with the CALIPSO overpass products. In section 4 we discuss our findings, and finally in section 5 we provide our conclusions and the future prospects of this study.

19

# 20 2 Data and methods

21 IRRA methodology is based on the remote sensing and in-situ measurement synergy, using 22 observations performed during the ACEMED campaign. Specifically, airborne active remote 23 sensing observations were performed with the Leosphere ALS450 lidar system acquiring 24 backscatter and depolarization profiles at 355 nm (Marenco et al., 2011; Chazette et al., 2012). The in-situ instruments (Table 1) included the TSI Integrating Nephelometer 3563 for the 25 26 measurement of the particle scattering coefficient, the Radiance Research Particle Soot 27 Absorption Photometer (PSAP) for retrieving the absorption coefficient, the Passive Cavity 28 Aerosol Spectrometer Probe 100-X (PCASP) and the 1.129 Grimm Technik Sky-Optical 29 Particle Counter (GRIMM) for retrieving particle number size distribution, as well as the 30 Aerodyne time-of-flight Aerosol Mass Spectrometer (AMS) for aerosol chemical composition. 31 Moreover, measurements of trace gases were acquired with the Chemical Ionization Mass Spectrometer (CIMS) and the fast fluorescence CO analyser, water vapour measurements were 32

provided by the Water Vapour Sensing System 2nd Generation (WVSS-II), along with temperature and pressure of the ambient air from the Rosemount deiced temperature sensor and the Reduced Vertical Separation Minimum system, respectively. More details about the instruments and measurements are given in section 2.2 with flight details given in section 3.1.

### 5 2.1 IRRA methodology for retrieving the ambient particle microphysics

IRRA characterizes ambient aerosol profiles by utilizing both the in-situ and remote sensing 6 7 data through an automated iterative scheme shown schematically in Fig. 1. In brief, the 8 measured dry particle parameters are "rehydrated" using the ISORROPIA II model to obtain 9 an estimate of the ambient particle size distribution and refractive index. Then, the dry particle 10 scattering and absorption coefficients, together with the ambient particle extinction and 11 backscatter coefficients, are calculated with the Mie theory (Mie, 1908; Bohren and Huffman, 12 1983). The retrieval is considered successful only if the calculations reproduce the airborne in-13 situ and lidar measurements; if this is not the case the input parameters are adjusted and the 14 process is repeated.

More specifically, for each straight level run (SLR) at a fixed altitude, the in-situ dry particle 15 size distribution and refractive index acquired from the PCASP, GRIMM and AMS 16 17 measurements are used in the retrieval as a first guess for the dry particle characteristics. Then, 18 the dry particle scattering and absorption coefficients are calculated using the Mie code of 19 Bohren and Huffman (1983), assuming spherical particles in the atmosphere, as indicated from 20 the low depolarization measurements acquired with the airborne lidar. The Mie calculations are 21 performed such as to reproduce the scattering and absorption coefficients measured by the in-22 situ optical instrumentation (i.e TSI nephelometer and PSAP), that are affected from inlet and 23 pipeline loses, as described in section 2.3.3. In order to optimize for these limitations, we use 24 at this stage a bimodal lognormal fit applied on the in-situ measurements (Eq. 1, red line in Fig. 25 2) truncated up to  $1.5 \,\mu\text{m}$  in radius (black dash line in Fig. 2).

$$\frac{dN}{dln(r)_{d}} = \frac{N_{fd}}{\sqrt{2\pi * ln(\sigma_{fd})}} \exp\left(-\frac{\left(ln(r) - ln(r_{mfd})\right)^{2}}{2ln(\sigma_{fd})^{2}}\right) + \frac{N_{cd}}{\sqrt{2\pi * ln(\sigma_{cd})}} \exp\left(-\frac{\left(ln(r) - ln(r_{mcd})\right)^{2}}{2ln(\sigma_{cd})^{2}}\right)$$
(1)

- 1  $\frac{dN}{dln(r)_d}$  is the dry particle number size distribution,  $N_{fd}$ ,  $N_{cd}$  are the total number concentrations,
- 2  $r_{mfd}, r_{mcd}$  are the geometric mean radii and  $\sigma_{fd}, \sigma_{cd}$  are the geometric standard deviation of 3 fine and coarse modes, respectively.
- Moreover, the dry particle refractive index is assumed to be spectrally constant and common for fine and coarse particles. That is because the information content in IRRA is not sufficient to resolve the refractive index spectral and size dependence. As a first guess we use the refractive index calculated from the in-situ chemical composition measurements, but this value is only an approximation and is expected to change, since the in-situ data do no provide a full chemical characterization of the particles.
- The next step, after defining the dry particle size distribution and refractive index, is to estimate the ambient particle properties by modelling their hygroscopic growth with ISORROPIA II model (a detailed model description is given in section 2.3.1). The ambient particle number size distribution is parameterized similarly to the dry particle number size distribution, considering that the geometric mean radius equals to the dry geometric mean radius multiplied by the hygroscopic growth factor  $f_a$  of the corresponding mode (Eq. 2, 3):

$$r_{mfa} = f_{gf} * r_{mfd} \tag{2}$$

$$r_{mca} = f_{gc} * r_{mcd} \tag{3}$$

16 The subscripts f and c denote the fine and coarse particle modes, respectively. The 17 corresponding  $f_g$  values are calculated from the water uptake predicted with ISORROPIA II. 18  $r_{mfa}$  and  $r_{mca}$  are the geometric mean radii of the modes. An example of an ambient size 19 distribution retrieval is shown in Fig. 2 (blue line) for RH=81%.

20 The real and imaginary parts of the ambient particle refractive index are calculated as following:

$$n_{af,c}(\lambda) = \left(1 - f_{wf,c}\right) * n_{df,c} + f_{wf,c} * n_w(\lambda) \tag{4}$$

$$k_{af,c}(\lambda) = \left(1 - f_{wf,c}\right) * k_{df,c} + f_{wf,c} * k_w(\lambda)$$
(5)

where  $n_{af,c}(\lambda)$  and  $k_{af,c}(\lambda)$  are the real and imaginary parts of the ambient refractive index,  $n_{af,c}$  and  $k_{af,c}$  are the same for dry particles,  $n_w(\lambda) + ik_w(\lambda)$  is the water refractive index,  $\lambda$ is the wavelength and  $f_{wf,c}$  are the water volume fractions in total volume of the ambient particles, provided by ISORROPIA II model.

1 Finally, IRRA aims to achieve a closure of the Mie-calculated optical properties of the ambient 2 and dry particles, with the lidar and in-situ measurements. These properties are the backscatter 3 and extinction coefficients at 355 nm calculated from the ambient properties, and the scattering 4 coefficients at 450, 550, 700 nm and absorption coefficient at 567 nm calculated from the dry 5 properties. The closure is achieved through the minimization of a cost function, using the Trust-6 Region-Reflective optimization algorithm (based on the interior-reflective Newton method 7 described in Coleman and Li, 1994; 1996) with the non-linear least-squares solver "lsqcurvefit" 8 of MATLAB. The cost function is the sum of the squares of the differences between the 9 measured and calculated optical properties, weighted by their "importance" for the retrieval, as 10 described in more detail in Appendix A. Briefly, starting from a first guess for the parameters 11 of the dry particle size distribution and refractive index, the optimization algorithm iteratively 12 searches the parameter space for a set that minimizes the cost function. The search stops after 13 different stopping criteria have been reached. For example, the cost function reduction is 14 smaller than the uncertainty of the measurements or the search step size is smaller than the 15 uncertainty of the parameter space (Dubovik, 2004). In our case these criteria cannot be strictly 16 quantified, due to inadequate information on measurement and parameter uncertainties, thus 17 the optimization procedure is set to stop after few ( $\sim 10$ ) iterations, after which there is no 18 considerable change in the cost function reduction or in the step size. The retrieval errors can 19 be quantified using the measurement uncertainties and the Jacobian matrix of the final 20 optimization step (Rodgers, 2000; Dubovik, 2004). Although they are not provided in the 21 current version of IRRA code, we plan to include them in the future versions.

22 2.2 Data

# 23 2.2.1 Airborne lidar

The airborne active remote sensing observations used in IRRA for the ACEMED campaign, were performed with the nadir-pointing Leosphere ALS450 lidar system, capable of acquiring particle backscatter and depolarization profiles at 355 nm (Marenco et al., 2011; Chazette et al., 2012). The measurements were acquired at night-flight, and the absence of daylight allowed the airborne lidar to measure with good signal to noise ratio (SNR). Lidar signals were measured with an integration time of 2 s and a vertical resolution of 1.5 m, and are smoothed vertically to a 45 m vertical resolution in order to improve SNR further. The vertical profiles of lidar

signals are then cloud-screened by eliminating those in the presence of clouds using the

- thresholds in Allen et al. (2014).
- The particle backscatter and extinction coefficients from the ALS450 system observations are
- calculated following the solution by Klett (1985), assuming a variable LR at 355 nm with
- height, and an aerosol extinction coefficient at 355 nm at a reference height in the far range.
- Both LR and reference extinction are calculated from the retrieved ambient size distribution
- and refractive index at each height.

# 8 2.2.2 Airborne in-situ

### 9 2.2.2.1. Particle drying from in-situ instruments

10 The inlets to the aircraft in-situ instruments dry the sampled air due to adiabatic compression 11 in the inlet during sampling, in addition to the cabin temperature and radiant heat from the lights 12 in the instruments. There is a chance this drying is only partial, with some residual water 13 remaining in the sample (e.g. Strapp et al., 1992; Snider and Petters, 2008). The partial drying is estimated from the instrument RH (and the particle chemical composition) and is taken into 14 15 account in modelling the particle hygroscopic growth with ISORROPIA II. Unfortunately, 16 instrument RH measurements are provided only for the nephelometer, with values ranging at 17 ~25-40%. We assume that these values are the same for PSAP. For PCASP and GRIMM optical 18 particle counter (OPC) measurements we consider a low RH of 30%, based on the work of 19 Strapp et al. (1992). Strapp et al (1992) indicate that particles with radius less than 5  $\mu$ m should 20 be dehydrated due to the residence time of 0.1-0.3 s in the low humidity environment of the 21 instrument. Even if this is not the case, the RH of 30% has a minor effect on particle hydration 22 for the samples analysed here, causing  $\sim 1\%$  growth in particle size. For the sake of simplicity 23 herein we call the partially dried particles as "dry particles".

#### 24 **2.2.2.2.** Particle size distribution measurements

The number size distributions were measured with PCASP and GRIMM OPCs. Both instruments measure the particle number size distribution by impinging light on the air sample and inferring the number and size of the particles from the light each particle scatters over a specified angular range (Rosenberg et al., 2012; Heim et al., 2008). PCASP operates a He-Ne laser at 0.6328 µm, measuring the particle scattering at 35-120° (primary angles) and 60° -145° (secondary angles), providing a (nominal) size range of 0.05–1.5 µm radius. GRIMM uses the

light of a laser diode at 0.683  $\mu$ m, measuring at 30 ° -150° (primary angles) and 81-99° 1 2 (secondary angles), providing a (nominal) size range of 0.125-16 µm radius. The number of 3 particles equals to the scattered light pulses, since each particle in the sample generates a light 4 pulse. The particle size is calculated comparing the height and width of the pulse with that from 5 calibration standards of known size distribution and refractive index, assuming that the sample 6 has the same refractive index as the calibration standard. This is the "nominal size" and the true 7 size can be then derived correcting for the particle refractive index, as described in Rosenberg 8 et al. (2012). For the PCASP we use the calibration standards from the Fennec 2011 campaign 9 (Rosenberg et al., 2012), and for the GRIMM we generate calibration standards assuming a bin 10 width uncertainty of 5%, based upon the manufacturers' specification. A detailed description 11 of handling and correcting the OPC size distribution data is provided in Appendix B.

The PCASP was wing-mounted on the BAe-146 aircraft, whereas the GRIMM was internally mounted and connected with a Rosemount inlet, thus sampled the air differently, through different inlets and pipelines. The effects of inlet efficiencies (enhancement/losses) and loses along the pipelines varied with altitude and ambient size distribution, affecting mainly the coarse mode particles (Ryder et al., 2013; Trembath et al., 2012). Inlet efficiency corrections are applied to PCASP using the methods of Belyaev and Levin (1974). The GRIMM OPC was not corrected for particle losses, and we expect the main loses to be for the largest particles.

19 As a validation of correctly handling the PCASP and GRIMM data, we compare the derived 20 PCASP and GRIMM number size distributions (after converting them to volume size 21 distributions) with the ambient volume size distributions provided by AERONET 22 measurements on the days before and after the BAe-146 aircraft night flight (Fig. 3). Note that 23 the AERONET does not provide vertically-resolved products, but rather the effective-column volume size distribution with units  $\mu m^3 \mu m^{-2}$ . For a direct comparison with PCASP and 24 GRIMM data (in  $\mu m^3 cm^{-3}$ ) we divide the AERONET size distribution with the aerosol layer 25 26 height (derived by the lidar measurements to be equal to ~3.5 km). The OPC data uncertainties 27 in the plot of Fig. 3 are calculated considering the refractive index uncertainty (Rosenberg et 28 al. 2012) and counting statistics (see Appendix B). For fine mode there is a very good agreement 29 among the two OPCs, but this is not the case for particles with radius  $>1.5 \mu m$ . The AERONET 30 volume size distributions are quite similar with the in-situ measurements for the fine mode, 31 with the AERONET particle volume (observations before the flight) to be within  $\sim \pm 60\%$  of the 32 PCASP and GRIMM particle volume for particles with radius