# Peer review of "Profiling aerosol optical, microphysical and hygroscopic"

_Atmospheric Measurement Techniques, 2016_

## Referee Comment (RC1) · Anonymous Referee #2 · 12 Aug 2016

Review of paper: Profiling aerosol optical, microphysical and hygroscopic properties in ambient conditions by combining in-situ and remote sensing, by A. Tsekeri, V. Amiridis, F. Marenco, A. Nenes, E. Marinou, S. Solomos, P. Rosenberg, J. Trembath, G. J. Nott, J. Allan, M. Le Breton, A. Bacak, H. Coe, C. Percival, and N. Mihalopoulos The authors present in details a complex procedure as IRRA-In-situ/Remote sensing aerosol Retrieval Algorithm. This combines airborne in-situ and lidar remote sensing data to retrieve vertical profiles of ambient aerosol optical, microphysical and hygroscopic properties, employing the ISORROPIA II model for acquiring the hygroscopic growth. The proposed methodology is improving our current understanding regarding humid conditions for aerosol characterization and is going to be valuable for applications in

aerosol-cloud interaction schemes. Also it could become a validation-tool for active space-borne sensors, as proven in here for the case of CALIPSO The text is very clear and even though is a long paper it reads well. I believe the manuscript is worth publication. Overall the paper provides useful information. The subject and the results of the paper will be of interest to lidar and atmospheric science researchers even though the presented techniques are not new.

---

## Referee Comment (RC2) · Anonymous Referee #3 · 12 Aug 2016

Profiling aerosol optical, microphysical and hygroscopic properties in ambient conditions by combining in-situ and remote sensing Alexandra Tsekeri et al.

Overall, this is an excellent manuscript where the authors have made a great job to convey all the information in an organized and clear way. All the sections and methods have been well supported and all the details have been referenced accordingly. The organization of the manuscript and the presentation of the data and results are of great quality. I consider this work is an important contribution for the atmospheric community as it provides important details on how attack the RH issue in remote sensing retrievals using the In-situ Remote sensing aerosol Algorithm (IRRA) and the ISOR-ROPIA II model to obtain the effect of RH on light scattering by aerosol particles. This

manuscript describes in detail the measurements, the calculations as well as assumptions considered for aerosol characterization. As it was very well expressed by the authors, this methodology offers a great number of applications in different aerosol scenarios where RH plays an important role and can even be used for the validation of actual remote sensing techniques. The authors also take account of different improvements for future work with IRRA, especially those of including polarization and non-spherical aerosol properties which will definitely help future airborne missions. In consequence, I consider this manuscript is appropriate for prompt publication in the Atmospheric Measurement Techniques journal.

---

## Author Comment (AC1) · 10 Oct 2016

Review of Referee #2:

Review of paper: Profiling aerosol optical, microphysical and hygroscopic properties in ambient conditions by combining in-situ and remote sensing, by A. Tsekeri, V. Amiridis, F. Marenco, A. Nenes, E. Marinou, S. Solomos, P. Rosenberg, J. Trembath, G. J. Nott, J. Allan, M. Le Breton, A. Bacak, H. Coe, C. Percival, and N. Mihalopoulos The authors present in details a complex procedure as IRRA-In-situ/Remote sensing aerosol Retrieval Algorithm. This combines airborne in-situ and lidar remote sensing data to retrieve vertical profiles of ambient aerosol optical, microphysical and hygroscopic properties, employing the ISORROPIA II model for acquiring the hygroscopic growth.

The proposed methodology is improving our current understanding regarding humid conditions for aerosol characterization and is going to be valuable for applications in aerosol-cloud interaction schemes. Also it could become a validation-tool for active space-borne sensors, as proven in here for the case of CALIPSO. The text is very clear and even though is a long paper it reads well. I believe the manuscript is worth publication. Overall the paper provides useful information. The subject and the results of the paper will be of interest to lidar and atmospheric science researchers even though the presented techniques are not new.

Answer to Referee #2:

We thank the reviewer for his/her comments.

The reviewer says that the presented techniques are not new, but we need to clarify here that the IRRA innovation does not lay on the individual techniques per se, but on their synergy in the retrieval. It is due to this synergy that IRRA manages to combine remote sensing with in-situ measurements in the retrieval, even for high relative humidity cases. To our knowledge, this has never been done before.

---

## Author Comment (AC2) · 10 Oct 2016

Review of Referee # 3:

Profiling aerosol optical, microphysical and hygroscopic properties in ambient conditions by combining in-situ and remote sensing Alexandra Tsekeri et al. Overall, this is an excellent manuscript where the authors have made a great job to convey all the information in an organized and clear way. All the sections and methods have been well supported and all the details have been referenced accordingly. The organization of the manuscript and the presentation of the data and results are of great quality. I consider this work is an important contribution for the atmospheric community as it provides important details on how attack the RH issue in remote sensing retrievals using the In-situ

[Figure]

Remote sensing aerosol Algorithm (IRRA) and the ISORROPIA II model to obtain the effect of RH on light scattering by aerosol particles. This manuscript describes in detail the measurements, the calculations as well as assumptions considered for aerosol characterization. As it was very well expressed by the authors, this methodology offers a great number of applications in different aerosol scenarios where RH plays an important role and can even be used for the validation of actual remote sensing techniques. The authors also take account of different improvements for future work with IRRA, especially those of including polarization and non-spherical aerosol properties which will definitely help future airborne missions. In consequence, I consider this manuscript is appropriate for prompt publication in the Atmospheric Measurement Techniques journal.

Answer to Referee #3:

We thank the reviewer for his/her very encouraging comments.

---

## Author Comment (AC3) · 10 Oct 2016

We thank the Editor and the Referees for the reviewing process.

Although both reviewers suggested the publication of the manuscript as is, we made some minor changes that finalize the text, without altering the concept of the presented work. We attach the new text.

Please also note the supplement to this comment:
http://www.atmos-meas-tech-discuss.net/amt-2016-193/amt-2016-193-AC3-supplement.pdf

[Figure]

[Figure]

**Supplement:**

**Profiling aerosol optical, microphysical and hygroscopic properties in ambient conditions by combining in-situ and remote sensing**

**Alexandra Tsekeri[1], Vassilis Amiridis[1], Franco Marenco[2], Athanasios Nenes[3,4,5], Eleni Marinou[1,6], Stavros Solomos[1], Phil Rosenberg[7], Jamie Trembath[8], Graeme J. Nott[8], James Allan[9,10], Michael Le Breton[9], Asan Bacak[9], Hugh Coe[9], Carl Percival[9], and Nikolaos Mihalopoulos[4]**

[1]{IAASARS, National Observatory of Athens, Athens, Greece}

[2]{Satellite Applications, Met Office, Exeter, UK}

[revised manuscript text omitted]

An alternative hybrid approach for obtaining well-constrained ambient aerosol profiles is through the synergy of active remote sensing observations with concurrent airborne in-situ measurements. To date, such efforts focused mostly on low-humidity profiles, where there is no difference between the ambient remote sensing measurements and the in-situ measurements performed under dry conditions inside the instruments (e.g. Weinzierl et. al, 2009High- humidity conditions have also been studied, but only for fine mode particle properties (e.g.

Ziemba et al. 2013), as the coarse particle hygroscopic growth is not as easily constrained with in-situ airborne techniques, mainly due to sampling inlet loses. The IRRA approach presented here addresses these limitations through the combination of in-situ and active remote sensing measurements with hygroscopic modelling, making possible the vertical profiling of fine and coarse particles even for humid conditions. For this purpose, the retrieval combines typical airborne in-situ instrumentation, measuring the dry particle size distribution and chemical composition, together with a simple backscatter lidar. The ambient remote sensing measurements are linked to the dry in-situ data through modelling of the particle hygroscopic growth with the ISORROPIA II model (e.g., Fountoukis and Nenes, 2007; Guo et al., 2015).

In the current study IRRA is applied on data collected in the framework of the EUFAR-ACEMED campaign ("evaluation of CALIPSO's **A**erosol **C**lassification scheme over **E**astern **MED**iterranean"), during which the FAAM BAe-146 research aircraft performed two under-flights of the Cloud–Aerosol Lidar and Infrared Pathfinder Satellite Observation (CALIPSO) satellite. The Mediterranean is considered ideal for the application and evaluation of IRRA retrieval scheme, as almost all globally-relevant aerosol types are encountered in the region, i.e. dust storms from desert or semi-arid areas in Africa, fresh and aged smoke from biomass burning, maritime aerosols, biogenic emissions, and anthropogenic aerosols (e.g. Lelieveld et al., 2002).

IRRA methodology is presented in section 2, along with a detailed description of the airborne in-situ and lidar measurements acquired during the ACEMED campaign, as well as ISORROPIA II and other models used. Section 3 presents the IRRA results for the ACEMED flight over Thessaloniki, Greece, along with a comparison with the CALIPSO overpass products. In section 4 we discuss our findings, and finally in section 5 we provide our conclusions and the future prospects of this study.

**2   Data and methods**

IRRA methodology is based on the airborne remote sensing and in-situ synergy, utilizing backscatter lidar measurements, along with the size distribution, chemical composition, scattering and absorption in-situ measurements. Specifically for the ACEMED campaign, airborne active remote sensing observations were performed with the Leosphere ALS450 lidar system acquiring backscatter and depolarization profiles at 355 nm (Marenco et al., 2011; Chazette et al., 2012); the in-situ instruments (Table 1) included the TSI Integrating Nephelometer 3563 for measuring the particle scattering coefficient at 450, 550 and 700 nm, the Radiance Research Particle Soot Absorption Photometer (PSAP) for the absorption coefficient at 567 nm, the Passive Cavity Aerosol Spectrometer Probe 100-X (PCASP) and the 1.129 Grimm Technik Sky-Optical Particle Counter (GRIMM) for measuring the particle number size distribution, and the Aerodyne time-of-flight Aerosol Mass Spectrometer (AMS) for providing the aerosol chemical composition. Moreover, measurements of trace gases were acquired with the Chemical Ionization Mass Spectrometer (CIMS) and the fast fluorescence CO analyser, water vapour measurements were provided by the Water Vapour Sensing System 2nd Generation (WVSS-II), along with temperature and pressure of the ambient air from the Rosemount deiced temperature sensor and the Reduced Vertical Separation Minimum system, respectively. More details about the instruments and measurements are given in section 2.2 with flight details given in section 3.1.

**2.1 IRRA methodology for retrieving the ambient particle microphysics**

IRRA characterizes the ambient aerosol profiles by utilizing both in-situ and remote sensing data through an automated iterative scheme shown schematically in Fig. 1. Briefly, the in-situ measured dry particle parameters are "rehydrated" using the ISORROPIA II model to obtain an estimate of the ambient particle size distribution and refractive index. Then, the dry particle scattering and absorption coefficients, together with the ambient particle extinction and backscatter coefficients, are calculated with the Mie theory (Mie, 1908; Bohren and Huffman, 1983). The retrieval is considered successful only if the calculations reproduce the airborne in-situ and lidar measurements; if this is not the case the input parameters are adjusted and the process is repeated.

For each straight level run (SLR) at a fixed altitude, the in-situ dry particle size distribution and refractive index acquired from the PCASP, GRIMM and AMS measurements are used in the retrieval as a first guess for the dry particle characteristics. Then, the dry particle scattering and absorption coefficients are reproduced using the Mie code of Bohren and Huffman (1983), assuming spherical particles in the atmosphere. The in-situ optical instrumentation (i.e TSI nephelometer and PSAP) are affected from inlet and pipeline loses, resulting in coarse particle undersampling, as described in section 2.3.3. For this reason, at this stage we consider a bimodal lognormal size distribution (Eq. 1, red line in Fig. 2) that is truncated up to 1.5 μm in radius (black dash line in Fig. 2).

[revised manuscript text omitted]

AERONET observations before the flight to be within ~±60% of the PCASP and GRIMM particle volume for particles with radius <1.5 μm. Similar results are shown in Haywood et al. (2003) for 0.1-1.0 μm radius range, for their fresh smoke aerosol plumes. For particles with radius >1.5 μm the agreement is worse, especially for GRIMM data, owing to the Rosemount inlet enhancement of the super-micron particles (as described in Trembath et al. (2012)). This is to be expected, since for sizes >1.5 μm the agreement between PCASP and GRIMM deteriorates as well. In any case, the comparison with AERONET data should be done with caution, since it refers to ambient particles, and the measurements are offset by ~9 hours. Moreover, differences can be also attributed to the AERONET size distribution cut-off at 15 μm. In general though, the good agreement for particles with radius <1.5 μm for all three datasets indicates that the airborne in-situ measurements of PCASP and GRIMM instruments provide trustworthy data for this size range, fitted to be used in our analysis.

**2.2.2.3. Chemical composition and refractive index**

The aerosol composition and mass distribution of volatile and semi-volatile components of aerosols as a function of particle size (with radius from 0.025 to 0.4 μm) were measured with the AMS instrument (Allan et al., 2003; Morgan et al., 2010; Athanasopoulou et al., 2015). AMS measures the mass loadings of the refractive aerosol fractions: sulphates, nitrates, ammonium, chloride and organics. Figure 4 shows the AMS measurements for the ACEMED case analysed here, indicating mixtures of inorganics/organics in the range of ~50/50 (the chloride mass concentration is very low and is not shown in the plot). Although the data refer mainly to fine mode particles, in our analysis we assume that they are representative of the coarse mode as well, since there are no measurements for the coarse particle chemical composition (the "coarse mode" denotes here to particles with radius >0.8 μm -see Fig. 3).

The chemical composition provided by the AMS can be used to estimate the particle refractive index, assuming that the particles are internally mixed and applying a volume mixing law to account for the contributions of the corresponding chemical groups (Highwood et al., 2012). For the calculations we need to consider a characteristic refractive index for each chemical group as well as a density to convert the AMS-measured dry mass to volume. Here we use the values provided in Highwood et al. (2012) (see Table 2): We assume the sulphate, nitrate and ammonium particles to be in the form of ammonium sulfate $((NH_4)_2SO_4)$ and ammonium nitrate $(NH_4NO_3)$, with density and refractive index provided by Toon (1976) and Weast (1985), respectively. For organics, we consider the properties of the organic carbon of the

Swannee River Fulvic Acid, as reported in Dinar et al. (2006) and Dinar et al. (2008). This approach is quite approximate, especially considering the refractive index variability of the "organics" group. In addition, the aerosol sampled is influenced by biomass burning (mainly due to high HCN and CO concentrations measured –see section 3.1) and may be strongly absorbing – this means that the uncertainty on the imaginary part is large. For these reasons the AMS-derived refractive index is used only as a first guess for the refractive index calculation with IRRA algorithm, as described in the methodology section 2.1. A similar approach was followed from McConnell et al. (2010), although focusing mainly on the imaginary part retrieval.

**2.2.2.4. Scattering and absorption**

The dry particle scattering coefficients at 450, 550 and 700 nm were measured on-board with the TSI Integrating Nephelometer 3563 and the absorption coefficient at 567 nm was measured with the PSAP (Orgen 2010). The scattering coefficient measurements are corrected for angular truncation, temperature and pressure (Anderson and Orgen 1998; Turnbull 2010). The absorption measurements are corrected for pressure, flow rate and spot size effects (Bond et al., 1999; Orgen 2010; Turnbull 2010). Both instruments were connected to modified Rosemount inlets (Trembath et al., 2012), suffering from inlet enhancement/losses as well as losses along the pipelines, and consequently did not measure the scattering properties of the whole size range of particles. For this reason, we consider a sampling cut-off for particles with radius >1.5 μm for the TSI nephelometer and PSAP measurements.

**2.2.2.5. Ambient relative humidity**

The ambient RH is estimated from the water vapour measurements from the WVSS-II instrument (Fleming and May, 2004). The WVSS-II uses a near-infrared tunable diode laser absorption spectrometer. Two WVSS-II instruments were mounted on the BAe-146 aircraft, sampling the air through the standard flush inlet and a modified Rosemount inlet, respectively. The water vapor measurements provided by the two instruments can be different (Vance et al., 2015), but for the case presented here the differences are small, of the order of less than 2% in ambient RH, thus what we used in our analysis is their average. The ambient RH calculation from the WVSS-II water vapour measurements is provided in Appendix C.

**2.3 Models**

**2.3.1 Hygroscopic growth model**

ISORROPIA II (Nenes et al., 1997; Fountoukis and Nenes, 2007) models the phase state and composition of aerosol composed of Na, $NH_4$, $NO_3$, Cl, $SO_4$, Mg, K, Ca and $H_2O$ in equilibrium with a gas phase composed of $NH_3$, $HNO_3$ and HCl. The model performance has been evaluated against comprehensive ambient datasets over a wide range of acidities, RH, and temperatures (Fountoukis and Nenes, 2006; 2007; Hennigan et al., 2015; Guo et al., 2015; Weber et al., 2016; Guo et al., 2016). In our analysis, we also consider the contribution of hygroscopic organics to the water uptake of the aerosol using the approach of Guo et al. (2015).

ISORROPIA II takes as input the aerosol precursor composition, along with the temperature, pressure and RH of the sample inside the instrument, and the temperature, pressure and RH of the ambient atmosphere, and it calculates the hygroscopic growth of fine and coarse modes. Since we assume the same chemical composition for fine and coarse particles, the hygroscopic growth is the same for both. The calculations involve various uncertainties, mainly from the hygroscopicity of the organic matter, the uncertainties and/or the variability in the RH measurements and the size-dependence of composition (that is not considered) within each mode and between modes.

Overall, ISORROPIA II provides an excellent estimation of the particle hygroscopic growth, especially at high RHs where the hydration has the greatest effect on the particle properties (e.g., Guo et al., 2015). The cumulative effect of particle composition on water uptake can be expressed using the hygroscopicity parameter $\kappa$ (Petters and Kreidenweis, 2007), derived from $f_g$ and RH as in Eq. 6:

$$\kappa = \frac{f_g{}^3 - 1}{RH/_{100 - RH}} \tag{6}$$

For mixtures of inorganics/organics in the range of ~50/50, as is the case here, the hygroscopicity parameter is 0.2 – 0.3 for RH>80% (Petters and Kreidenweis, 2007; Chang et al., 2010; Mikhailov et al., 2013). Airborne measurements performed above the Aegean Sea during the Aegean-Game campaign (Bezantakos et al., 2013), which was coupled with ACEMED, showed similar values for $\kappa$. Over multiple years, measurements of particle hygroscopicity at the South Aegean site of Finokalia, Crete also exhibit very similar values (Bougiatioti et al., 2009; 2011; 2016; Kalkavouras et al., 2016). ISORROPIA II retrieves

$\kappa \approx 0.25$ for RH > 80%, and lower values for smaller RHs (see Fig. 5). Given that the hygroscopic growth data reported by Bezantakos et al., (2013) corresponds to RH>80%, this consistency between predictions and observations is a strong indication that the internal mixture assumption applies, and that the AMS composition data is representative of the ambient aerosol. Moreover, the drop in predicted hygroscopicity for RH<80% is consistent with observed behaviour for aerosol particles (e.g. Guo et al., 2015).

**2.3.2 Source-receptor simulations**

In order to investigate the origin of the aerosol plumes in the scene analysed here, a number of backward and forward Lagrangian simulations of particle dispersion are performed with FLEXPART-WRF model (Brioude et al., 2013). These simulations are driven by WRF_ARW (Skamarock et al., 2008) hourly fields at 4×4 km horizontal resolution. Initial and boundary conditions for the WRF model are from the National Centers for Environmental Prediction (NCEP) final analysis (FNL) product at 1°×1° resolution. The sea surface temperature (SST) is the daily NCEP SST analysis at 0.5°×0.5° resolution. Furthermore, in order to derive information of smoke dispersion for the forward runs, fire hot spots are obtained from the Moderate Resolution Imaging Spectroradiometer (MODIS) Fire Information for Resource Management System (FIRMS) database.

**2.4 CALIPSO product**

The derived ambient particle properties during the ACEMED campaign are used to evaluate the CALIPSO products. CALIPSO satellite carries CALIOP, an elastic backscatter lidar operating at 532 and 1064 nm, equipped with a depolarization channel at 532 nm, delivering global vertical profiles of aerosols and clouds. The CALIPSO Level-2 (L2) aerosol layer products used in the current study are generated by automated algorithms and provide a description of the aerosol layers in respect to horizontal and vertical extend, along with the particle backscatter and extinction coefficient profiles. A detailed description of the L2 algorithms is provided in Vaugan et al. (2004) and Winker et al. (2009).

The CALIPSO Vertical Feature Mask (VFM) L2 product (Vaughan et al. 2004), classifies aerosols and clouds based on their optical properties and external information of geographical location, surface type and season (Omar et al, 2005; 2009). The classification scheme differentiates six subtypes of aerosol particles: polluted continental, smoke, dust, polluted dust, clean marine and clean continental. An example of the attenuated backscatter coefficient and the associated VFM classification, for the case analysed here, is shown in Fig. 7b and c. Burton et al. (2013) have validated the CALIPSO classification scheme using collocated airborne High Spectral Resolution Lidar (HSRL) measurements during 109 CALIPSO under-flights and reported a relatively trustworthy classification for mineral dust (80%) which falls to 62% for marine particles, 54% for polluted continental, 35% for polluted dust and only 13% for smoke.

**3   Results**

**3.1   ACEMED flight overview**

The scope of the ACEMED EUFAR campaign was the evaluation of the CALIPSO aerosol classification scheme using high quality airborne aerosol measurements along with ground-based lidar, sunphotometric and in-situ observations. ACEMED was clustered with the Aegean-Game (**Aegean** Pollution: **G**aseous and **A**erosol airborne **ME**asurements) campaign (Tombrou et al., 2015), held from 31 August to 9 September 2011 with the FAAM (http://www.faam.ac.uk/) BAe-146 research aircraft, based in the island of Crete (Greece). Two CALIPSO under-flights were performed for ACEMED, on 2 September and during the night between 8 and 9 September. Here only the latter flight is considered (FAAM flight B644, Fig. 6), due to higher SNR lidar measurements during the night.

The BAe-146 aircraft approached Thessaloniki area from the Southeast, flying at an altitude of ~5 km above mean sea level. Once in the operating area, it flew on a SSW to NNE transect back and forth between 40N and 41.2N, sampling at different altitudes over both land and ocean (coastline at 40.6N, see Fig. 6). A first SLR was done at 5.1 km altitude, in order to fly above the aerosol layers so as to provide full profiles with the use of the on-board lidar. Then, the aircraft flew a series of SLRs at altitudes 3.9, 3.2, 2.7, 2.1, 1.8 and 1.3 km, and in each of these, data have been collected with the in-situ instrumentation. The aircraft then profiled the atmosphere, returning to high level (4.8 km) for an additional remote sensing survey in the shape of a box pattern around the sampling area. The lidar measurements used in the current analysis were acquired when the aircraft was flying at 5.1 and 4.8 km, and the in-situ measurements were acquired at 3.2, 2.7, 2.1, 1.8 and 1.3 km (at 5 and 3.9 km the in-situ data showed no presence of particles).

Figure 7 shows the vertical profiling of the atmosphere along the flight, as depicted in the range-corrected backscatter signal at 355 nm from the airborne lidar (Fig. 7a), and the curtain of the attenuated backscatter coefficient as this is provided by CALIPSO L1 product at 532 nm (Fig. 7b). The airborne lidar depolarization measurements were low (not shown here), indicating spherical particles in the scene. In both Fig. 7a and b there is strong indication of cloud formation at ~3 km in part of the flight above land (shown as white features). Large RHs have also been observed in the airborne WVSS-II RH measurements in Fig. 8 at that height, where the cloudy parts above land show RHs of 92-98%. These cloudy parts are excluded from the CALIPSO aerosol subtype VFM product (see Fig. 7c) and the corresponding lidar vertical profiles are excluded from our analysis. At the cloud-free parts the RH is higher above land (80-90% at 2-3.5 km and 60% below 2 km) and lower above ocean (70-80% at 2-3 km and <60% below 2 km) (Fig. 8).

FLEXPART source-receptor simulations show the advection of smoke from biomass burning towards the region of interest in Fig. 9. The wind direction over the Balkans was mainly NW. However due to the complex topography at the area and the development of low-level thermal circulations along the coastlines (sea-breeze), the wind pattern at the lowest 1 km in the troposphere was rapidly changing with time, affecting also the dispersion of smoke. Such wind channelling and sea-breeze formation is adequately resolved in the finer WRF grids. The emission sensitivity (residence time) for a 24-hour backwards simulation and for two representative locations (one over land and one over ocean) is shown in Fig. 9a. The red triangles denote the position of the active fires during this period as obtained by the MODIS fire product (https://earthdata.nasa.gov/earth-observation-data/near-real-time/firms). These results identify six hot spots that fall within the emission sensitivity area and so are most likely responsible for the smoke transport over the region of interest.

Taking into account the positions and times of detection of the six emission points we perform a forward simulation of smoke dispersion, assuming constant emission rates of 0.15 kg s$^{-1}$ and constant smoke injection heights at 1 km. The vertical cross section of smoke total particulate matter (TPM) is shown in Fig. 9b (the location of the cross section is indicated by the dashed black line in Fig. 9a). In order to compensate for the possible time lags in modelled smoke transport we compute the average TPM concentration for the period 00:00-02:00 UTC from the corresponding 30-minute model outputs (i.e., 00:00, 00:30, 01:00, 01:30 and 02:00 UTC). Figure 9b shows elevated smoke plumes over the northern land part at about 2-3 km and near the surface (the latter though being below the FAAM BAe-146 flight level). The results indicate also the presence of a lower (1-2.5 km) smoke plume over the ocean. The elevated smoke plumes above the southern land part in Fig. 9b are out of the FAAM BAe-146 flight range.

The smoke presence above Thessaloniki is also supported by the biomass burning proxies HCN and CO measurements, acquired with CIMS (Le Breton et al., 2013) and the fast fluorescence CO analyser (Gerbig et al., 1999), respectively. The HCN is used as a biomass burning tracer (Lobert et al., 1990) since its lifetime in the smoke plume can potentially exceed several weeks. As indicated in Le Breton et al. (2013), HCN concentrations higher than six standard deviations from the median background concentration are highly correlated with CO concentrations indicating biomass burning plumes. Indeed, the HCN concentrations seem to exceed the smoke plume detection threshold at altitudes from 2 to 3.5 km (Fig. 10). These values are strongly correlated with the corresponding CO concentrations, with a correlation of $R^2 = 0.8$ (not shown here), strongly indicating the smoke presence. The measurements agree well with source-receptor simulations in Fig. 9, except for the lower part of the smoke plume above the ocean, which is not depicted in the HCN data.

Although the CALIPSO L2 aerosol classification product identifies the smoke over Thessaloniki city (at the land part of the flight), it seems that at the southern part of the scene, above ocean, the algorithm misclassifies the layers almost completely (Fig. 7c). We believe that this is partly due to the different classification criteria for smoke above land versus above ocean, as these are defined by Omar et al. (2009) for the CALIPSO classification scheme. More specifically, as can be seen in Fig. 2 in Omar et al. (2009) the non-depolarizing aerosol plumes are classified as smoke plumes above ocean only if they are "elevated layers" (a layer is considered "elevated" when the layer base > 2.5 km, or if 0.5 km < base < 2.5 km then top > 4km or depth > 2 km; J. Tackett, personal communication). More analysis on the CALIPSO "hard limit" that can be potentially imposed on the aerosol classification at coastal areas due to the different land/ocean classification criteria, can be found in the work of Kanitz et al. (2014). Due to this discontinuity we decided to perform our analysis at the land and ocean parts separately, in order to examine the possible differences present in CALIPSO L2 product. For the ocean retrieval we use the area from 40 to 40.6 degrees latitude (marked with the light blue rectangle in Fig. 7c), whereas for land we use only the two cloud-free 5-km segments (corresponding to CALIPSO L2 5-km-profiles) indicated with the orange rectangle in Fig. 7c, in the area from 40.6 to 41.2 degrees latitude.

**3.2  Flight above land**

Using the combination of airborne in-situ and active remote sensing measurements with the IRRA retrieval scheme described in section 2.1, we retrieve profiles of the ambient particle properties above land and ocean. For the retrieval above land we use the lidar measurements taken at 5 km and the in-situ measurements acquired during the SLRs at 3.2, 2.7 and 1.8 km. The comparisons between the measured and calculated dry and ambient particle optical properties show both excellent agreement (Fig. 11 and Table D1 in Appendix D), with the relative differences to be below 5 %. The only exception is the lidar extinction coefficient at 1.8 km, with ~10 % relative difference. This may be due to the incoherence of the lidar and in-situ measurements there, due to temporal variability of the atmospheric properties, with the lidar measurements to be an average of the flight segment at ~5 km between 00:20 and 00:27 UTC, and the in-situ measurements at 1.8 km to refer at 01:38 to 01:42 UTC (see Fig. 6b).

Overall, as seen in Fig. 11 and 12 the very high RHs that exceed 90% at 3.2 km and 80% at 2.7 km (see Fig. 8), have a large hydration effect on the ambient particle optical and microphysical properties. Figure 11 shows quite vividly the hydration effect on the ambient backscatter and extinction coefficients at 355 nm, at 2.7 and 3.2 km, whereas at 1.8 km the effect is small. The comparison of dry (red dots) with ambient calculations (blue dots) for the backscatter and extinction coefficients, highlight the deficiency of dry in-situ measurements to reproduce the ambient particle optical properties in humid conditions.

A similar conclusion can be drawn from the retrieved ambient (number and volume) size distributions provided in Fig. 12 and Table 3, and the respective refractive indices in Table 4. The hydration effect of both fine and coarse modes is notable, especially for the high-RH layers at 2.7 and 3.2 km, with a water content of 55% and 80% of the total volume, respectively. The retrieved dry particle fine mode is well-fitted to the measured PCASP and GRIMM data, with 95% of the calculated size distribution data points to be within two error bars of the measured data. For the coarse mode the fit is also acceptable, although the high uncertainty in the in-situ measurements does not allow a definite conclusion.

The retrieved ambient lidar ratio at 355 nm of ~70-90 sr and the dry SSA at 550 nm of ~0.9-0.95 (Fig. 11) indicate the presence of absorbing particles along the flight path, in good agreement with the source-receptor simulations, as well as with the HCN and CO airborne in-situ measurements, all showing the advection of smoke over Thessaloniki area. The retrieved geometric mean radius and standard deviation of the fine mode are similar to measurements detailed in Johnson et al. (2016) for the SAMBBA, DABEX and SAFARI-2000 campaigns for aged smoke. The retrieved dry particle refractive indices of 1.54-1.6 + i0.008-0.021 are within the range of typical values for biomass burning particles, and the corresponding ambient refractive indices (1.38-1.55 + i 0.002-0.019) are close to the AERONET 8-year global aerosol climatology of Dubovik et al. (2002). More specifically, Dubovik et al. (2002) report a range of 1.47±0.03 to 1.52±0.01 for the real part and 0.00093±0.003 to 0.021±0.004 for the imaginary part.

**3.3  Flight above ocean**

For the retrieval above ocean we use the airborne lidar measurements at 5 km and the in-situ measurements from the SLRs at 3.2, 2.7, 2.1 and 1.3 km. The calculated optical properties reproduce well the measurements, as shown in Fig. 13 (and in Table D2 of Appendix D), with most of the relative differences to be below 15%. For the lidar backscatter and extinction coefficients at the lower SLRs at 2.1 and 1.3 km these differences are larger and range at ~30-100%. As explained for the retrieval above land as well, this may be due to the temporal variability of the atmosphere, resulting in the lidar seeing a different aerosol plume than the in-situ measurements, especially for the lower SLRs (see Fig. 6b).

The results support the presence of smoke mixed with other aerosol types (e.g. urban pollution), with the ambient lidar ratio at 355 nm to be ~55-75 sr and the dry particle SSA at 550 nm to be ~0.9-0.95. Figure 14 and Table 5 show the retrieved (number and volume) size distributions of dry and ambient particles, at different altitudes and Table 6 shows the corresponding refractive indices. As with the land retrieval, the fine mode PCASP and GRIMM measurements are well-fitted, whereas for the coarse mode the uncertainty is higher. The hydration effect is mostly obvious at 3.2 km (RH of ~80% with 40% water content in the particle total volume), whereas it is very small at 1.3 km (RH at ~55%).

The retrieved geometric mean radius and standard deviation of the fine mode are smaller than the values reported in Johnson et al., (2016) indicating mixing with finer aerosol (e.g. urban pollution). The retrieved dry refractive indices of ~1.50-1.66+ i0.01-0.019 have similar values with the retrieved refractive indices above land, although the real part of 1.66 at 2.7 km is higher. Moreover, the ambient refractive index values of ~1.48-1.6+ i0.006-0.015 are comparable to AERONET climatological values (Dubovik et al., 2002), indicating the smoke particle presence above the ocean as well.

**3.4 Comparison with CALIPSO L2 product**

Using the retrieved ambient size distribution and refractive index at different altitudes we calculate the ambient backscatter, extinction coefficient and lidar ratio at 532 nm and compare them with the corresponding CALIPSO L2 products. Above land, the smoke layer at ~2-3.5 km is correctly identified by the CALIPSO aerosol classification scheme (Fig. 7c), and a prescribed LR at 532 nm of 70 sr (assigned for smoke particles) is used for the CALIPSO L2 backscatter and extinction coefficient retrievals. Figure 15 presents the results for the above-land retrieval, showing good agreement with the CALIPSO L2 product. The small differences seen are within the spatial variability and can be due to the time difference of CALIPSO overpass (at 00:30 UTC) and the longer FAAM BAe-146 flight (at 00:05-01:45 UTC). The LRs at 532 nm calculated with the retrieved ambient size distributions and refractive indices are 70-80 sr, supporting the presence of the smoke particles. The optical properties are calculated also using the dry particle size distribution and refractive index (red circles in Fig. 15) to highlight the problems that arise when using dry in-situ measurements for satellite validation for cases of high RH.

Over the ocean the retrieved ambient LRs at 532 nm at ~60-75 sr are lower than over land, indicating smoke particles mixed with other aerosol types. CALIPSO detects the aerosols correctly, but does not classify them as smoke (except only for one 5-km profile), and as shown in Fig. 7c, it classifies the particles either as polluted dust (LR=65 sr), or polluted continental (LR=70 sr) or marine aerosol (LR=20 sr), resulting in variable and lower LRs (25-70 sr). The mean LR is close to 60 sr, thus the mean backscatter and extinction coefficients are not excessively affected by this misclassification. The CALIPSO misclassification is due to the constraint applied in the algorithm to identify only the elevated layers as smoke layers above the ocean.

**3.5 Scattering growth factor**

The enhancement of aerosol scattering due to the hygroscopic growth is shown in Fig. 17 with the scattering growth factor at 532 nm. The scattering growth factor is the ratio of the ambient aerosol scattering coefficient, to the dry aerosol scattering coefficient. Figure 17 shows that the scattering at RH=94% is almost 4 times larger than in the dry state. These values fall within the range of Köhler curves for aged smoke particles and can be used in climate models for the estimation of hydrated aged smoke particle scattering (e.g. Johnson et al., 2016).

**4 Discussion**

The results presented here are very encouraging for the IRRA retrieval scheme performance. First, IRRA succeeds to reproduce both dry in-situ and ambient remote sensing measurements, even in humid conditions of RH>80-90%, by considering both dry and ambient particle states in the retrieval scheme and by effectively modelling the particle hygroscopic growth the ISORROPIA II model. Second, IRRA manages to provide the complete set of the particle microphysical properties, overcoming the deficiencies in the in-situ measurements due to the insufficient coarse mode size distribution and chemical composition sampling. We do not claim that the coarse mode retrieval is highly accurate with IRRA, but at least it closely reproduces the measurements and provides similar results to the climatological values of biomass burning particles, for the smoke plume case we analysed here. A more complete set of inputs, as in-situ coarse mode sampling and multi-wavelength lidar measurements, should increase the retrieval input information content and provide more accurate results. Third, IRRA retrieval is not gravely affected by possible uncertainties in the in-situ measured microphysical properties, since these are only used as a first guess in the iterative retrieval scheme. The unknown coarse mode chemical composition is an exception, since it directly affects the estimation of the coarse mode hygroscopic growth in the ISORROPIA II model.

**5 Conclusions**

IRRA utilizes successfully the airborne active remote sensing and in-situ measurements in order to provide a consistent characterization of the ambient aerosol at different altitudes, using typical airborne instruments employed by the FAAM BAe-146 aircraft flight. The retrieved ambient properties found to be mostly consistent also with the collocated CALIPSO L2 product. Specifically, smoke plumes are identified along the flight path, which are detected from CALIPSO classification scheme above land, but not above ocean.

One of the main shortfalls of the case analysed here is the large uncertainty in the airborne in-situ measurements regarding the coarse particle size distribution and chemical composition. In future IRRA applications, in-situ particle sizers achieving high accuracy measurements of the coarse mode should be employed, along with filter sampling of the coarse particles. We should note though that despite of the limited coarse mode information, our retrieval provides plausible results for the coarse particles as well, for the case presented here.

The achievement of IRRA is the overall successful profiling of the ambient aerosol microphysical, optical and hygroscopic properties utilizing the combination of dry particle property measurements, active remote sensing and ISORROPIA II hygroscopic growth modelling, all in one retrieval scheme. The potential of IRRA lies beyond the case study analysed here, providing an effective aerosol characterization in ambient conditions of high importance for aerosol/cloud interaction, radiative transfer and climate studies.

We should highlight that IRRA is optimized with the measurement set acquired during the ACEMED campaign, but this is not a limiting factor of its applicability. The basic concept of combining vertically-resolved in-situ and active remote sensing measurements can be satisfied using a different measurement strategy as well. For example, after applying minor changes, IRRA can combine vertically-resolved in-situ measurements from aircraft or less-costly unmanned aerial vehicle (UAV) platforms, with ground-based or satellite lidar measurements.

IRRA can also be developed further in the future, to include ground-based or airborne polarization measurements which should help to decrease the aerosol retrieval uncertainty, especially for the particle refractive index (Mishchenko and Travis, 1997). Another feature we plan to implement is the retrieval of non-spherical particle properties, employing non-spherical particle scattering codes in the algorithm (e.g. the T-matrix code as in Dubovik et al. (2006), or the Advanced Discrete Dipole Approximation as in Gasteiger et al. (2011)). This will extend the applicability of IRRA to dust particle characterization as well.

For the application presented here, it has been shown that it is feasible with IRRA to evaluate space-borne profiling measurements. Beyond CALIPSO, IRRA can be further applied for the validation of the new NASA CATS mission but also future ESA missions like ADM-Aeolus and EarthCARE.

**Appendix A: IRRA optimization retrieval scheme**

IRRA retrieval methodology shown in Fig. 1 is automated utilizing the non-linear least-squares solver "lsqcurvefit" of MATLAB. The lsqcurvefit calculates the dry and ambient particle size distributions and refractive indices that minimize the difference between the calculated and measured optical properties in a least-squares sense (Eq. A1).

$$min\|F(SD_d, m_d, f_{gf,c}, f_{wf,c}) - y\|_2^2 = min \sum_i \left(F(SD_d, m_d, f_{gf,c}, f_{wf,c})_i - y_i\right)^2 \qquad (A1)$$

$$(A2)$$

$$y = \{sc_{450}, sc_{550}, sc_{700}, abs_{565}, \alpha_{355}, \beta_{355}, NC_{0.8}, NC_{1.1}\}$$

$$F(SD_d, m_d, f_{gf,c}, f_{wf,c}) \qquad (A3)$$
$$= \left\{ \begin{array}{c} F_{sc_{450}}(SD_d, m_d), F_{sc_{550}}(SD_d, m_d), F_{sc_{700}}(SD_d, m_d), F_{absc_{565}}(SD_d, m_d), \\ F_{\alpha_{355}}(SD_d, m_d, f_{gf,c}, f_{wf,c}), F_{\beta_{355}}(SD_d, m_d, f_{gf,c}, f_{wf,c}), F_{NC_{0.8}}(SD_d), F_{NC_{1.1}}(SD_d) \end{array} \right\}$$

$$(A4)$$

$$SD_d = \{r_{mfd}, \sigma_{fd}, N_{fd}, r_{mcd}, \sigma_{cd}, N_{cd}\}$$

The retrieval is performed for each height separately. In Eq. A1, the summation over "$i$" denotes
the different optical properties, $y$ is the vector of the measured optical properties (Eq. A2) and
$F$ is the vector of the calculated optical properties (Eq. A3). $SD_d$ is the vector of the dry size
distribution parameters (Eq. A4), $m_d$ is the dry particle refractive index and $f_{gf,c}, f_{wf,c}$ are the
hygroscopic growth and water volume fraction of fine and coarse ambient particles. The
retrieved parameters are the $SD_d$ and $m_d$, whereas $f_{gf,c}, f_{wf,c}$ provided by ISORROPIA II.

The $y$ vector contains the in-situ measurements of the scattering coefficients at 450, 550 and
700 ($sc_{450}, sc_{550}$ and $sc_{700}$, respectively), the absorption coefficient at 565 nm ($abs_{565}$), as
well as the lidar measurements of extinction ($\alpha_{355}$) and backscatter coefficient at 355 nm ($\beta_{355}$).
In order for the retrieval not to be under-constrained, with less measurements than retrieved
parameters, $y$ also contains the in-situ measured number concentration of dry particles at 0.8
and 1.1 μm ($NC_{0.8}$ and $NC_{1.1}$). We use these coarse particle concentration values to constrain
more effectively the coarse mode retrieval, for which the in-situ measurements provide
accepted accuracy for sizes <1.5 μm (radius) (see discussion in Section 2.2.2).

$F$ vector contains the corresponding calculated values of $y$: $F_{sc_{450}}, F_{sc_{550}}, F_{sc_{700}}$ and $F_{absc_{565}}$ are
the scattering coefficients at 450, 550 and 700 nm and the absorption coefficient at 565 nm,
calculated from the dry particle number size distribution ($SD_d$) and refractive index ($m_d$), utilizing Mie scattering calculations. Moreover, $F_{\alpha_{355}}$ and $F_{\beta_{355}}$ are the extinction and backscatter coefficients at 355 nm, calculated from the ambient number size distribution (derived from $SD_d$ and $f_{gf,c}$, as in Eq. 2, 3) and refractive index (derived from $m_d$ and $f_{wf,c}$, as in Eq. 4, 5), with Mie scattering calculations. Finally, $F_{NC_{0.8}}$ and $F_{NC_{1.1}}$ are the values of $SD_d$ at 0.8 and 1.1 μm.

The lsqcurvefit function employs the Trust-Region-Reflective optimization algorithm (based on the interior-reflective Newton method described in Coleman and Li, 1994; 1996) to minimize the cost function in Eq. A1. For the first iteration the parameters $SD_d$ and $m_d$ are set equal to a first guess, derived from the in-situ measurements. Subsequently, the algorithm searches for a set of parameters that minimizes the cost function. The minimization is done using a simpler function (defined by the first two terms of the Taylor approximation of the cost function) which models reasonably well the cost function behaviour in a "trust region" around the parameter set. A trial step is then computed by minimizing the modelled function. If the cost function is minimized as well, then the parameter set is updated using the trial step, and the trust region is expanded. Otherwise, the parameter set remains unchanged, the trust region is shrunk and the trial computation is repeated. The optimization procedure stops after predefined stopping criteria are reached. These may include the minimum cost function value, the minimum size of the trial step or a maximum number of iterations. The first two criteria are defined from the input measurement and the retrieved parameter uncertainties, respectively, which are not available for the current analysis. Thus, for the case analyzed here, we used a maximum number of iterations as the stopping criterion.

Moreover, the algorithm has the capability to use constrains for the lower and upper bounds of the retrieved parameters. We utilize this feature for the dry particle fine and coarse mode parameters, so as the retrieved parameters are not very different than the in-situ measurements. The dry particle refractive index is also constrained, so as to be within realistic values, with the real part from 1.3 to 1.7 and the imaginary part from 0 to 0.1. These values cover well the range of values provided from the worldwide aerosol climatology from 8-year AERONET data by Dubovik et al. (2002).

Last, the fitted parameters of $y$ do not have all the same importance for our retrieval. More specifically, we are not interested in reproducing with high accuracy the number concentration measurements at 0.8 and 1.1 μm ($NC_{0.8}$ and $NC_{1.1}$), or at least not as much as the measured optical properties. For this reason we "weight" the fit, by first normalizing to 1 each parameter in $y$ (dividing it with its value) and then multipling with a weight that is a measure of the importance of the parameter fitting. The same multiplication factors are applied on the parameters of $F$ vector. For the case analysed here we used weights of 1 for the optical properties and of 0.1 for the number concentrations at 0.8 and 1.1 μm. The "weighting" of the fit can be very useful in the general case of combining measurements of different accuracies and it has been used in other retrievals in the literature (e.g. Dubovik and King, 2000). The weights should be derived based on the measurement accuracy, but if this is not easy to define (as is the case here), even qualitative numbers of "more" or "less" confidence in the measurements can help the retrieval.

**Appendix B: Size distribution data handling and calibration**

The number size distribution data from PCASP and GRIMM instruments come in the form of number of particles, per $cm^3$, per size bin. The number concentration for each size bin is normalized by $dln(r_{max}) - dln(r_{min})$ ($r_{min}$ and $r_{max}$ refer to the minimum and maximum bin radius, respectively) to get the log-normal number size distribution ${dN}/{dln(r)}$. The log-normal volume size distribution ${dV}/{dln(r)}$ is then calculated by multiplying ${dN}/{dln(r)}$ with the volume of the particles in each bin.

The data are also inspected for spurious values, using the associated counting error, which for each size bin is defined as the inverse square root of the number of particles in the bin. The data associated with counting errors larger than 0.3 (corresponding to less than three particles in the size bin) are screened out. Moreover, the data are corrected for the refractive index assumption using the true refractive index and calibration standards, with the mieconscat and the cstodconverter software (http://sourceforge.net/projects/mieconscat/ and http://sourceforge.net/projects/cstodconverter/, respectively), as described in Rosenberg et al.

(2012For this correction we assume that the particles are homogeneous and spherical. The uncertainty for the bin width is provided from the cstodconverter software and the uncertainty in the volume of the bin is calculated using the uncertainty in the bin width and the counting uncertainty of each bin.

**Appendix C: RH calculation**

The ambient RH is calculated from the WVSS-II water vapour volume mixing ratio ($WV_{VMR}$)

measurements and the ambient pressure ($P$) measurements as following:

$$RH = \frac{WV_{VMR}\, P}{e} 100$$

(C1)

where the $WV_{VMR}$ is in $m^3 m^{-3}$, $P$ is in $hPa$, $e$ (in $hPa$) is the vapour pressure of water calculated from the temperature ($T$) measurements (in $C$) as in Lowe and Ficke (1974):

$$e = a_0 + T\left(a_1 + T\left(a_2 + T\left(a_3 + T\left(a_4 + T(a_5 + a_6 T)\right)\right)\right)\right)$$

(C2)

with $a_0 = 6.107799961$, $a_1 = 4.436518521\ 10^{-1}$, $a_2 = 1.428945805\ 10^{-2}$, $a_3 =$

$2.650648471\ 10^{-4}$, $a_4 = 3.031240396\ 10^{-6}$, $a_5 = 2.034080948\ 10^{-8}$ and $a_6 =$

$6.136820929\ 10^{-11}$.

**Appendix D: Measured and retrieved optical properties**

Table D1. Measured versus calculated (bold) in-situ measurements of the dry particle scattering coefficient at 450, 550 and 700 nm and SSA at 550 nm, and remote sensing measurements of the ambient backscatter and extinction coefficients at 355 nm, above land. The spatial (horizontal) variability of the measurements is provided as the standard deviation around the mean value.

| | Airborne in-situ | | | | Airborne remote sensing | |
|---|---|---|---|---|---|---|
| Height (km) | Scattering coefficient at 450 nm (km$^{-1}$) | Scattering coefficient at 550 nm (km$^{-1}$) | Scattering coefficient at 700 nm (km$^{-1}$) | SSA at 550 nm | Backscatter coefficient at 355 nm (km$^{-1}$) | Extinction coefficient at 355 nm (km$^{-1}$) |
| 3.2 | 0.076±0.002 | 0.054±0.002 | 0.032±0.002 | 0.95±0.01 | 0.004 | 0.310 |

| Height (km) | | | | | | |
|---|---|---|---|---|---|---|
| | **0.074** | **0.054** | **0.034** | **0.95** | **0.004** | **0.307** |
| 2.7 | 0.082±0.004 | 0.055±0.003 | 0.033±0.002 | 0.91±0.01 | 0.002 | 0.192 |
| | **0.080** | **0.056** | **0.033** | **0.91** | **0.002** | **0.200** |
| 1.8 | 0.071±0.004 | 0.051±0.002 | 0.031±0.002 | 0.90±0.01 | 0.001 | 0.099 |
| | **0.070** | **0.051** | **0.031** | **0.90** | **0.001** | **0.108** |

Table D2. As for Table D1, for the retrieval above ocean.

| | Airborne in-situ | | | | Airborne remote sensing | |
|---|---|---|---|---|---|---|
| Height (km) | Scattering coefficient at 450 nm (km$^{-1}$) | Scattering coefficient at 550 nm (km$^{-1}$) | Scattering coefficient at 700 nm (km$^{-1}$) | SSA at 550 nm | Backscatter coefficient at 355 nm (km$^{-1}$) | Extinction coefficient at 355 nm (km$^{-1}$) |
| 3.2 | 0.070±0.011 | 0.049±0.008 | 0.030±0.005 | 0.93±0.03 | 0.003 | 0.151 |
| | **0.072** | **0.053** | **0.033** | **0.94** | **0.003** | **0.144** |
| 2.7 | 0.070±0.017 | 0.050±0.012 | 0.030±0.008 | 0.91±0.02 | 0.002 | 0.111 |
| | **0.077** | **0.056** | **0.035** | **0.92** | **0.002** | **0.121** |
| 2.1 | 0.083±0.007 | 0.060±0.005 | 0.038±0.004 | 0.91±0.01 | 0.003 | 0.155 |
| | **0.089** | **0.067** | **0.044** | **0.92** | **0.002** | **0.153** |
| 1.3 | 0.116±0.005 | 0.085±0.004 | 0.053±0.003 | 0.92±0.01 | 0.001 | 0.089 |
| | **0.110** | **0.086** | **0.058** | **0.93** | **0.002** | **0.168** |

**Acknowledgements**

The research leading to these results received funding from the European Community's Seventh
Framework Programme (FP7/2007-2013) under grant agreement n°227159 (EUFAR: European

Facility for Airborne Research in Environmental and Geo-sciences) and the UK Natural

Environment Research Council [Grant ref: NE/E018092/1]. Airborne data was obtained using the BAe-146-301 Atmospheric Research Aircraft [ARA] flown by Directflight Ltd and managed by the Facility for Airborne Atmospheric Measurements [FAAM], which is a joint entity of the Natural Environment Research Council [NERC] and the Met Office. This research has received funding from the European Union's Horizon 2020 research and innovation programme under grant agreement No 654109. The publication was supported by the

European Union Seventh Framework Programme (FP7-REGPOT-2012-2013-1), in the framework of the project BEYOND, under grant agreement no. 316210 (BEYOND– Building

Capacity for a Centre of Excellence for EO-based monitoring of Natural Disasters).

Athanasios Nenes acknowledges support from a Georgia Power Faculty Scholar Chair and a

Cullen-Peck Faculty Fellowship.

The authors would also like to acknowledge the contribution of Jim Haywood, Alan Vance and

Kate Turnbull from the UK Met Office, Jason Tackett from the CALIPSO Lidar Science

Working Group at NASA Langley Research Center and Ioannis Binietoglou from IAASARS,

National Observatory of Athens.

Table 1. The in-situ instruments and data acquired from the FAAM BAe-146 research aircraft during the ACEMED campaign.

| Property measured | Instrument | Important information about the data |
|---|---|---|
| Dry aerosol number size distribution | Passive cavity aerosol spectrometer probe 100-X (PCASP) | Nominal size range: 0.05 - 1.5 μm (radius) |
| | 1.129 GRIMM Technik Sky-Optical Particle Counter (GRIMM) | Nominal size range: 0.125 – 16 μm (radius) |
| Dry aerosol chemical composition and mass | Aerodyne time-of-flight aerosol mass spectrometer (AMS) | Nominal size range: 0.025 - 0.4 μm (radius) |
| Dry aerosol light scattering coefficient at 450, 550 and 700 nm | TSI Integrating Nephelometer 3563 (Nephelometer) | We consider a sampling cut-off at 1.5 μm (radius) |
| Dry aerosol light absorption coefficient at 567 nm | Radiance Research Particle Soot Absorption Photometer (PSAP) | We consider a sampling cut-off at 1.5 μm (radius) |
| HCN | Chemical Ionization Mass Spectrometer (CIMS) | - |
| CO | Fast fluorescence CO analyser | - |
| Water vapor volume mixing ratio | Water Vapor Sensing System version two (WVSS-II) | - |
| Air temperature | Rosemount deiced temperature sensor | - |
| Static air pressure | Reduced Vertical Separation Minimum system | - |

Table 2. Refractive indices and densities used for the refractive index calculation from AMS

data acquired from the FAAM BAe-146 research aircraft during the ACEMED campaign.

| Chemical species | Refractive index at 550 nm | Density ($g\ cm^{-3}$) | References |
|---|---|---|---|
| Ammonium Sulphate $(NH_4)_2SO_4$ | 1.53-0i | 1.77 | Toon (1976) |
| Ammonium Nitrate $NH_4NO_3$ | 1.611-0i | 1.8 | Weast (1985) |
| Organic carbon of the Swannee River Fulvic Acid | 1.538-0.02i | 1.5 | Dinar et al. (2006) Dinar et al. (2008) |

Table 3. Retrieved number size distribution parameters of dry and ambient particles, for the retrieval above land. The total number concentration and geometric standard deviation is the same for dry and ambient particles.

| | Dry particles | | | Ambient particles | | |
|---|---|---|---|---|---|---|
| Height (km) | Total number concentrations $N_{fd}, N_{cd}$ $(cm^{-3})$ | Geometric mean radii $r_{mfd}, r_{mcd}$ (μm) | Geometric standard deviations $\sigma_{fd}, \sigma_{cd}$ | Total number concentrations $N_{fa}, N_{ca}$ $(cm^{-3})$ | Geometric mean radii $r_{mfa}, r_{mca}$ (μm) | Geometric standard deviations $\sigma_{fa}, \sigma_{ca}$ |
| 3.2 | 778, 0.7 | 0.1, 0.7 | 1.5, 1.6 | 778, 0.7 | 0.2, 1.1 | 1.5, 1.6 |
| 2.7 | 1317, 0.9 | 0.1, 0.5 | 1.5, 1.9 | 1317, 0.9 | 0.1, 0.7 | 1.5, 1.9 |
| 1.8 | 726, 0.8 | 0.1, 0.4 | 1.4, 1.9 | 726, 0.8 | 0.1, 0.5 | 1.4, 1.9 |

Table 4. Retrieved refractive indices of dry and ambient particles, for the retrieval above land.

| Height (km) | Retrieved refractive index | |
|---|---|---|
| | Dry particles | Ambient particles |
| 3.2 | 1.54+i0.008 | 1.38+ i0.002 |
| 2.7 | 1.60+i0.018 | 1.46+ i0.008 |
| 1.8 | 1.58+i0.021 | 1.55+i0.019 |

Table 5. Same as Table 3, for the retrieval above ocean.

| Height (km) | Dry particles | | | Ambient particles | | |
|---|---|---|---|---|---|---|
| | Total number concentrations $N_{fd}, N_{cd}$ $(cm^{-3})$ | Geometric mean radii $r_{mfd}, r_{mcd}$ (μm) | Geometric standard deviations $\sigma_{fd}, \sigma_{cd}$ | Total number concentrations $N_{fa}, N_{ca}$ $(cm^{-3})$ | Geometric mean radii $r_{mfa}, r_{mca}$ (μm) | Geometric standard deviations $\sigma_{fa}, \sigma_{ca}$ |
| 3.2 | 2814, 0.2 | 0.05, 1.6 | 1.8, 1.4 | 2814, 0.2 | 0.06, 1.9 | 1.8, 1.4 |
| 2.7 | 1500, 0.6 | 0.08, 0.6 | 1.5, 2.4 | 1500, 0.6 | 0.08, 0.6 | 1.5, 2.4 |
| 2.1 | 1833, 0.6 | 0.08, 1.3 | 1.6, 1.8 | 1833, 0.6 | 0.08, 1.4 | 1.6, 1.8 |
| 1.3 | 1427, 0.4 | 0.1, 1.1 | 1.6, 1.6 | 1427, 0.4 | 0.1, 1.1 | 1.6, 1.6 |

Table 6. Same as Table 4, for the retrieval above ocean.

| Height (km) | Retrieved refractive index | |
|---|---|---|
| | Dry particles | Ambient particles |
| 3.2 | 1.59+i0.01 | 1.48+i0.006 |
| 2.7 | 1.66+i0.019 | 1.6+i0.015 |
| 2.1 | 1.59+i0.015 | 1.56+i0.013 |
| 1.3 | 1.50+i0.015 | 1.50+i0.014 |

[Figure]

Figure 1. IRRA iterative retrieval scheme used for the estimation of the ambient particle microphysical property profiles, based on the airborne in-situ and remote sensing measurements available during the ACEMED campaign, and the hygroscopic growth modelling of

ISORROPIA II.

[Figure]

Figure 2: Number size distributions used for the aerosol optical property calculations. The red line denotes the bimodal lognormal fit on the measurements , the black dash line the truncated size distribution used to model the dry in-situ measured scattering and absorption coefficients, and the blue line the size distribution used to model the ambient backscatter and extinction coefficient lidar measurements. The measured in-situ number size distributions are denoted with pink and light blue dots, for PCASP and GRIMM OPC data, respectively. The data are acquired at 2.7 km above Thessaloniki, on 9 September 2011, at 01:04-01:12 UTC, during the ACEMED campaign.

[Figure]

Figure 3. The measured dry volume size distributions from PCASP (pink) and GRIMM (light blue),  acquired at 2.7 km above Thessaloniki, on 9 September 2011, at 01:04-01:12 UTC. The vertical error bars denote the volume uncertainty estimates, and the horizontal error bars the bin width uncertainties. The black lines indicate the column ambient size distributions from

AERONET before (at September 8, 2011, 15:28 UTC, denoted with filled circles) and after the flight (at September 9, 2011, 08:25 UTC, denoted with open circles).

[Figure]

Figure 4. The measured dry mass concentrations from AMS for organics, sulphate, ammonium
and nitrate, acquired during the ACEMED campaign, above Thessaloniki, on 9 September
2011, at 00:05-01:45 UTC. The error bars denote the horizontal variability on each SLR.

[Figure]

Figure 5. Hygroscopicity parameter calculated with ISORROPIA II for the flight above
Thessaloniki, Greece, on September 9, 2011, during the ACEMED campaign.

[Figure]

a)                                                      b)

Figure 6. a) The FAAM BAe-146 aircraft flight track above Thessaloniki, Greece, on

September 9, 2011, at 00:05-01:50 UTC (green line) and the CALIPSO track at 00:30 UTC

(red dots). b) The FAAM BAe-146 flight latitude-altitude profile (green line). The flight segments used in the current analysis are denoted with orange colour above land and with light blue colour above ocean.

[Figure]

             a)

             b)

             c)

Figure 7. a) The Leosphere ALS450 lidar range-corrected signal at 355 nm, for the FAAM

BAe-146 flight, above Thessaloniki, Greece, on September 9, 2011, at 00:05-00:27 UTC (the white line separates the ocean and land parts, at 40.6 N latitude), b) the CALIPSO attenuated backscatter coefficient at 532 nm, and c) the CALIPSO aerosol subtypes (VFM), for the

CALIPSO overpass at 00:30 UTC. The light blue and orange rectangles mark the area used to compare with the FAAM BAe-146 flight measurements above ocean and land, respectively.

[Figure]

Figure 8. Averaged RH measurements from the WVSS-II instrument, above land (orange circles for cloud-free area and pink circles for cloudy area) and ocean (light blue circles), during the FAAM BAe-146 aircraft flight above Thessaloniki, Greece, on September 9, 2011, at 00:48-

01:50 UTC.

[Figure]

a)                                                b)

Figure 9. a) 24-hour emission sensitivity (in logarithmic scale) [s m³ kg⁻¹] for particles that originate from the first 2.5 km of the FLEXPART-WRF model and are observed on 9

September 2011, 00:30 UTC at heights between 1-4 km above land and ocean, at Thessaloniki area. The red triangles indicate MODIS hot spot locations during that period. b) Cross-section of two-hour average concentration of smoke TPM (µg m⁻³) predicted with the dispersion model forward simulations along the FAAM-BAe-146 flight, on September 9, 2011, 00:00-02:00

UTC, indicated with the dashed black line in (a).

[Figure]

Figure 10. HCN concentration during the FAAM BAe-146 flight above Thessaloniki, Greece, on September 9, 2011, at 00:48-01:50 UTC. The data are marked for the flight path above land (orange circles) and ocean (light blue circles). The black line at 280 ppt marks the biomass burning plume threshold detection, equal to six standard deviations of the median background

HCN concentration (Le Breton et al., 2013).

[Figure]

Figure 11. Airborne in-situ and remote sensing optical properties, along with the corresponding calculated optical properties. From left to right: scattering coefficients at 450, 550 and 700 nm from TSI nephelometer (blue, green and red stars for measurements and dots for calculations), single scattering albedo (SSA) at 550 nm from PSAP and TSI nephelometer (black stars for measurements and dots for calculations), backscatter and extinction coefficients at 355 nm (blue line) retrieved from the lidar measurements, along with the corresponding calculated optical properties for dry and ambient particles (red and dark blue dots, respectively), and the calculated lidar ratio at 355 nm for dry and ambient particles. The data refer to the flight segment above land, above Thessaloniki, Greece, on September 9, 2011, at 00:20-01:42 UTC. The error bars in the first two plots denote the spatial variability of the measurements during each SLR, rather than instrumental uncertainty. The calculated optical properties corresponding to the in-situ measurements are calculated with truncated size distributions at 1.5 μm, whereas for the remote sensing calculations the size distributions are not truncated.

[Figure]

Figure 12. Retrieved number (left) and volume (right) size distributions along with the airborne in-situ measurements from PCASP and GRIMM OPCs at the altitudes of 1.8, 2.7 and 3.2 km. The red line denotes the dry particles and the blue line the ambient particles. The PCASP and GRIMM size distributions are truncated at 1.5 μm, showing the effect of the inlets in the sampled volume. The data refer to the flight segment above land, above Thessaloniki, Greece, on September 9, 2011, at 00:56-01:42 UTC.

[Figure]

Figure 13. As for Fig. 11, for the flight segment above ocean, above Thessaloniki, on September

9, 2011, at 00:06-01:50 UTC.

[Figure]

Figure 14. As for Fig. 12, for the flight segment above ocean, above Thessaloniki, on
September 9, 2011, at 00:45-01:50 UTC.

[Figure]

Figure 15. Backscatter (left), extinction (middle) and LR (right) at 532 nm, calculated from the retrieved ambient particle properties of FAAM BAe-146 flight above land (dark blue circles), and provided by the CALIPSO L2 product (green line) for the CALIPSO overpass above

Thessaloniki, Greece, on September 9, 2011, at 00:30 UTC. The errorbars in CALIPSO profiles denote the spatial variability and not the uncertainty of the CALIPSO L2 product. The calculated dry particle optical properties are also shown with red circles.

[Figure]

Figure 16. As for Fig.15, for the flight segment above the ocean.

[Figure]

Figure 17. The scattering growth factor at 532 nm, acquired from the retrieved aerosol microphysical properties during the ACEMED campaign, above Thessaloniki, on 9 September

2011.